# Evaluating the Contribution of Lean Construction to Achieving Sustainable Development Goals

Sada Hasan [1,2], Zeynep Işık [1] and Gökhan Demirdöğen [1,*]

1   Department of Civil Engineering, Yildiz Technical University, Davutpaşa Caddesi, Istanbul 34220, Turkey; sada.hasan@std.yildiz.edu.tr (S.H.); zeynep@yildiz.edu.tr (Z.I.)
2   Department of Civil Engineering, Al-Qadisiyah University, Ad Diwaniyah P.O. Box 1881, Iraq
*   Correspondence: gokhand@yildiz.edu.tr

**Abstract:** The construction industry is scrutinized and criticized for its impact on environmental degradation. Nowadays, while the lean construction philosophy and Sustainable Development Goals (SDGs) aim to alleviate the adverse environmental effects of the construction industry, their synergies remain unclear and ambiguous. Therefore, this study aims to explore the synergies between lean construction principles and the Sustainable Development Goals (SDGs) and their combined efficacy in mitigating the environmental footprint of the construction industry. In the study, a comprehensive three-step methodology, involving a literature review, focus group discussions, and quantitative Delphi technique analysis, was employed. The analysis uncovers that SDGs (ensuring clean water and sanitation, promoting clean energy, fostering economic growth and decent work, improving infrastructure and innovation, building sustainable cities and communities, promoting responsible consumption and production, conserving oceans, and preserving terrestrial ecosystems) have 63 extremely important linkages and 251 very important linkages with lean construction principles. The analysis results indicate that the synergies are categorized under economic (39%), environmental (42%), and social (19%) factors. Moreover, the strategic triad of lean principles, i.e., "Reducing non-value-adding, focusing on all processes, and continuous improvement", emerged as key in fostering extremely important interactions. This study's novelty lies in its integrating of Koskela's lean principles with the 17 SDGs and 169 targets of Agenda 2030, offering strategic insights for aligning construction processes with the broader 2030 agenda for enhanced sustainability in the construction industry. The findings contribute to finding out the how lean construction principles serve the SDGs.

**Keywords:** SDGs; lean principles; construction industry; Delphi technique

## 1. Introduction

The construction industry consumes about half of the raw materials extracted in the world [1]. The diverse range of materials necessary for construction projects, such as aggregates, cement, steel, and wood, fuels this demand. Furthermore, the industry's involvement in infrastructure development further drives resource extraction. This heavy reliance on raw materials strains global resource reserves and contributes to environmental degradation and increased greenhouse gas emissions. The construction sector consumes about 36% of the final energy consumption globally [1,2]. Furthermore, the construction and demolition process also constitutes about 25–32% of the total waste generated in the European Union [2]. Therefore, increasing awareness of ecologically friendly production techniques and resource-efficient methodologies is crucial to reducing the environmental impact of the construction industry [3]. Moreover, construction enterprises face a compelling need to adopt sustainable technologies and strategies [4,5] and promote responsible resource utilization. Consequently, these companies strive for enhanced competitiveness, considering factors like globalization, climate change, resource scarcity, and social and environmental responsibility [6].

Sustainable Development Goals (SDGs) represent an international initiative to reduce the environmental impact of business operations, which applies to industries like construction. The SDGs consist of 17 main targets (including 169 goals) focussed on addressing poverty, inequality, and bias, determined in 2015 [7–10]. The construction industry has a significant role in achieving the goals of the 2030 agenda for the SDGs due to its significant resource and energy consumption [11–13]. While Sustainable Development Goals (SDGs) are aimed at reducing human-induced impacts on the environment, economy, and society, lean principles and techniques in the construction process align with these goals [14,15]. Consequently, both the SDGs and lean principles aspire to advance sustainable development while safeguarding the needs of future generations [9,16].

While construction activities that focus on sustainability practices help to achieve SDGs, lean construction techniques are among the techniques that reduce the environmental impact of the construction industry by eliminating waste. In other words, the lean construction philosophy serves as another approach to gaining a competitive edge, aiming at waste elimination and optimal resource utilization [6,17,18]. It seeks to elevate efficiency, cost-effectiveness, quality, and customer value, with a focus on worker well-being, health and safety, and communication [19,20].

The review of the literature suggests that lean principles strongly intersect with environmental sustainability, which is evident in waste reduction, the promotion of value-added activities, the minimization of material waste, reduced working hours, and heightened safety measures [21,22].

While the studies conducted by researchers delve into critical aspects of lean construction and sustainability, they largely overlook the integration of Sustainable Development Goals (SDGs) into their analyses. For example, De Carvalho et al. focused on examining integrative synergies between lean principles and sustainability throughout a building's lifecycle, yet they did not explicitly consider how these synergies align with the broader objectives outlined in the SDGs [14]. Similarly, Li et al. evaluated the implementation of lean construction in Chinese firms but did not explore the potential alignment of lean practices with SDGs or their contributions to sustainable development beyond organizational efficiency [23]. Additionally, Carvajal-Arango et al.'s [15] study investigated the positive impacts of lean practices on sustainability during the construction phase, yet they did not assess these impacts in the context of SDGs or explore how lean principles could contribute to achieving specific SDG targets [15]. Therefore, while these studies provide valuable insights into the intersections of lean construction and sustainability, they fall short of explicitly addressing the broader sustainability agenda outlined by the SDGs. This research aims to bridge this gap by incorporating SDGs into the analysis and exploring how lean practices can contribute to achieving sustainable development on a broader scale. In addition, the lack of discovery of a synergy between lean principles and SDGs hinders lean and sustainable actions, as well as the measurement of the performance of these two concepts specific to the construction industry [24].

Therefore, the purpose of the study is to find and evaluate the synergies between lean principles in construction and SDGs to achieve the goals of Agenda 2030. As to our methodology, first, a literature review was conducted to develop an interaction matrix. The interaction matrix was evaluated by experts via a focus group discussion (FGD) session. With the use of an FGD session, validation of the interaction and the identification of new interactions that had not been considered in the literature were enabled. Finally, the Delphi technique was employed to calculate the importance level of the synergies quantitatively. The research question given below will be answered throughout the study:

Research Question: How do Sustainable Development Goals (SDGs) and lean construction principles interact synergistically to reduce the environmental impact of the construction industry?

The contribution of this study is its providing of a new perspective, not widely explored in previous studies, to better see the direct and indirect contribution of buildings to the SDGs. Also, this study focuses on the synergies between SDGs and lean construction

principles that have not been discovered yet. This helps to synthesize the effect of lean construction principles in achieving SDGs. Therefore, the researchers can uncover synergistic opportunities by considering their complementation. Moreover, the adoption of SDGs can be facilitated by using the existing theory (lean construction principles). Since lean construction principles are implemented in industries, there is more awareness about this concept, which may serve to enhance SDG-related efforts in academia and public/private institutions. Furthermore, the two concepts require intensive collaboration between policymakers, industry practitioners, researchers, etc. Construction companies and policymakers can use the study results to mitigate the environmental impact of the construction industry.

The present study has been structured as follows: (i) the "Introduction" presents background knowledge and the study's rationale; (ii) the "Literature Review" supports the theoretical background of the integrated concepts; (iii) the "Methodology" section explains the methodological steps taken to discover synergies between lean construction and SDGs; (iv) the "Results" section presents the outcome of the focus group discussion and Delphi study; (v) the "Discussion" section summarizes the reasons behind the outcomes and their implications; (vi) "The Contribution of the Study" section summarizes the theoretical and practical contributions of the study; and (vii) "Conclusions" concludes the paper.

This study was conducted specifically for the construction industry. While awareness of construction is high, knowledge of Sustainable Development Goals (SDGs) is relatively low in comparison. To address this issue, experienced academicians and practitioners were selected to explore the synergies between the two concepts. However, providing more examples for SDGs can help to reveal the impact of lean construction in terms of SDGs.

## 2. Literature Review

### 2.1. The Implementation of Lean Principles in the Construction Industry

Lean management, an inventive strategy pioneered by the Toyota Motor Company, focuses on optimizing customer value, minimizing production waste, and maintaining high quality standards [25].

One of the strategies employed in the construction industry to minimize production costs and enhance profitability stems from the lean thinking philosophy. This approach emphasizes continuous improvement, offering a strategic advantage by eliminating non-value-adding activities in the production process to reduce costs [26].

In the construction sector, lean principles involve 11 principles that are paramount for efficiency and quality [27–30]. Foremost, the principle of reducing non-value-adding activities involves trimming downtime and resource-intensive tasks that fail to benefit the end customer [19,27–35]. The principle of prioritizing customer needs is fundamental, ensuring that the construction process adds tangible value for them [27,31,32,35]. The principle of streamlining operations, minimizing diversity and uncertainty in various processes, is crucial, establishing clear standards and values from the outset [27–29,32,36,37]. The principle of cycle time reduction hinges on eliminating unnecessary activities and variations, spanning processing, control, waiting, and motion times [27–29,33,34,36,38]. The principle of simplification is key, whether it pertains to processes, components, or materials, as it cuts costs and enhances productivity [27,28,32,39]. The principle of flexibility in production output is achieved through swift adjustments, maintaining a tight alignment between production and demand [27,32]. The principle of transparency is vital, achieved through meticulous monitoring and dissemination of information using visual aids, reducing errors along the way [28,36]. All processes must be scrutinized and facilitated by a robust monitoring and control system [28,36]. The principle of continuous improvement offers a strategic advantage by eliminating non-value-adding activities in the production process to reduce costs [21,26,28,32,36,38,40,41]. The principles of workflow analysis and optimization are routine, ensuring maximum efficiency before implementation [27,41] Lastly, scrutinizing weaknesses and strengths allows for targeted enhancements, perpetuating the cycle of improvement within construction endeavors [27]. These principles collectively underpin lean construction, driving towards excellence in performance and client satisfaction.

In the literature, many researchers have dealt with the principles of lean construction that lead to sustainability through waste reduction, the promotion of value-added activities, decreased material waste, shortened working hours, and increased safety [4,21,39,42–46]. Other studies have focused on the relationship between green certification systems and lean techniques for high-rise projects [47]. Also, studies have examined the synergies, benefits, and trade-offs between lean thinking and sustainability in the construction industry [15,47–50] presented the synergies between 11 lean construction principles and 11 sustainability criteria. However, the literature lacks studies explicitly monitoring the relationship between the SDGs and targets according to Agenda 2030 and the principles of lean construction. Table 1 gives the lean construction principles used in the study.

**Table 1.** The lean construction principles used in the study.

| Code | L1 | L2 | L3 | L4 | L5 | L6 | L7 | L8 | L9 | L10 | L11 |
|---|---|---|---|---|---|---|---|---|---|---|---|
| Lean Construction Principles | Reducing non-value adding activities | Focus on customer needs | Reducing diversity and uncertainty in processes | Reducing cycle time | Simplifying processes, components, materials | Increasing production output flexibility | Increasing the transparency of production processes | Focus on all processes. | Integrating continuous improvement into processes | Analyze and optimize workflows before they change | Comparison for weakness and superiority detection |

### 2.2. The Employment of Sustainable Development Goals (SDGs) in the Construction Industry

The United Nations report [51,52] was adopted by 193 states at the United Nations General Assembly. This has provided a globally agreed sustainable development framework consisting of 17 goals and 169 targets to be achieved by 2030. But progress toward the 2030 targets is perilously slow, especially for the most disadvantaged and marginalized groups [53].

The Sustainable Development Goals consist of 17 main goals and 169 specific goals. These goals encompass eradicating poverty, ensuring food security, promoting health and well-being, achieving quality education, fostering gender equality, ensuring clean water and sanitation, promoting clean energy, fostering economic growth and decent work, improving infrastructure and innovation, reducing inequalities, building sustainable cities and communities, promoting responsible consumption and production, addressing climate action, conserving oceans and terrestrial ecosystems, promoting peace and justice, and strengthening global partnerships. Because some goals are not expressed as specific numbers, the United Nations has also developed a framework of 232 indicators to monitor and review the goals. These goals, accompanied by their targets and indicators, provide a comprehensive roadmap for collective action towards a more sustainable and equitable future for all. The indicator framework will be refined annually and reviewed comprehensively by the Statistical Commission at its fifty-sixth session, to be held in 2025.

Previous literature reviews have explored the implementation of lean principles in the construction industry, emphasizing their significance for efficiency and quality. These principles prioritize reducing non-value-adding activities, streamlining operations, and fostering continuous improvement. While many researchers have highlighted the benefits of lean construction for sustainability, they often overlook the explicit integration of SDGs in their analyses. Similarly, while the construction industry is guided by the SDGs framework adopted by 193 states, progress towards these goals has been slow. Despite the comprehensive nature of the SDGs and the accompanying indicators developed by the United Nations, existing studies often fail to explicitly consider the alignment between lean construction principles and the SDGs, and there is a notable absence of explicit integration between them in the literature. This gap highlights the need for research to explore how lean practices can contribute to achieving sustainable development targets outlined in the SDGs.

### 3. Methodology

A three-step approach was employed to fulfill research objectives, as depicted in Figure 1. After defining the research problem and establishing the research objectives, we moved on to discovering the synergy between lean construction and SDGs. The first step, the literature review, was performed to develop an interaction matrix. The

second step includes a focus group discussion (FGD) session to (i) validate the interaction between lean construction and SDGs discovered in the literature review and (ii) identify new interactions between lean construction and SDGs not found in the literature. In the latter step, identified interactions were quantitatively assessed by experts via the Delphi technique. Thus, the impact of interaction was categorized to present the lean principles most related to the SDGs.

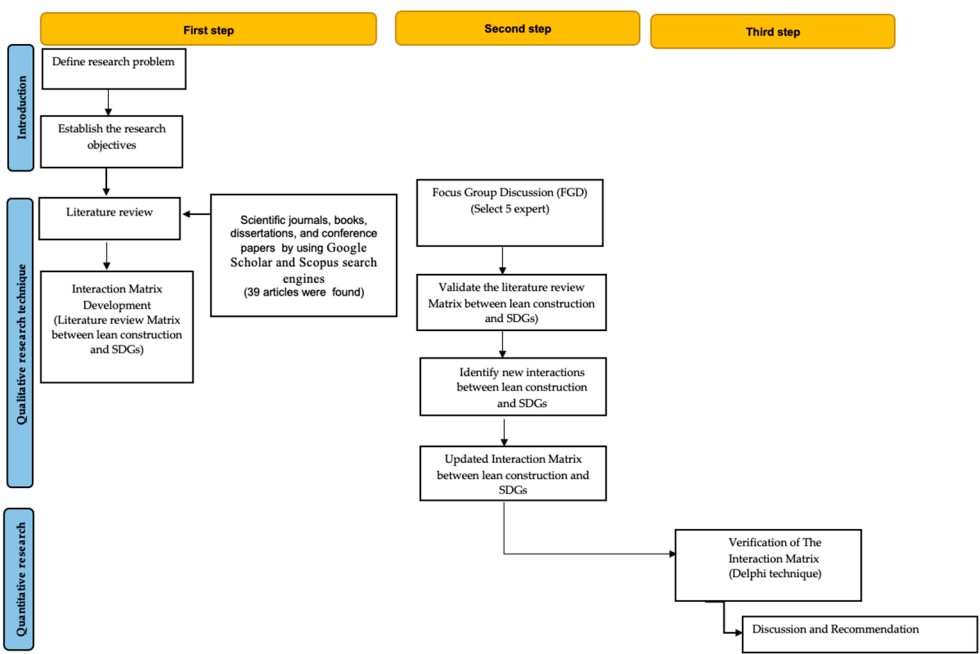

**Figure 1.** Research methodology.

The literature review aimed to identify studies addressing specific areas of interest [54] and establish connections between various research topics, a well-established method of investigation of SDGs in construction. For instance, Ogunmakinde et al. (2022) [55] employed this approach to explore the contributions of the circular economy to SDGs [55]. The review was unfolded, where relevant keywords were identified, as shown in Figure 2. They were searched using reputable databases—Google Scholar, Scopus, and Web of Science—known for their rigorous scrutiny of construction-related literature [56,57]. By using the time frame of 2015–2023, we aligned our study with the introduction of SDGs in 2015. As a primary result of the research, we found that no sources directly dealt with the relationship between lean construction principles and Sustainable Development Goals (SDGs). Rather, they dealt with lean principles and how they achieve sustainability elements, without diagnosing any goal specifically within Agenda 2030.

As may as 17 SDGs were tested with 169 goals and their relationship with lean principles was examined. This process yielded a total of 68 articles. In the subsequent review phase, the duplicates (18 articles) were eliminated, and the abstracts of the remaining 50 articles were scrutinized to identify their relevance to the objectives. After a comprehensive assessment of the abstracts, 39 articles were identified, encompassing conference papers and journal articles. Subsequently, we synthesized and integrated the findings from these articles into our study's discourse according to the linkage strategy of the lean principle with SDGs, as shown in Figure 2.

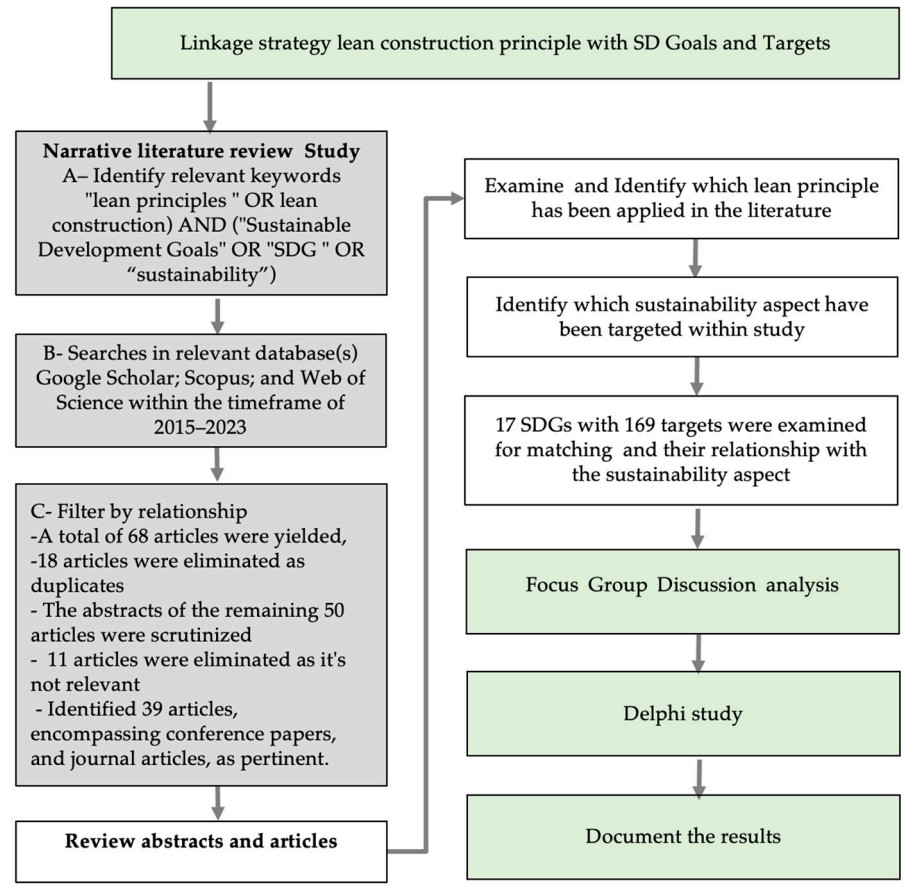

**Figure 2.** Linkage strategy of the principle of lean construction with SDGs.

A focus group discussion (FGD), a qualitative technique, involves deliberate participant selection for in-depth problem exploration, particularly in uncharted professional domains, necessitating expert involvement for accurate data collection [58]. Each FGD comprised 4 to 21 participants [58–61] utilized FGDs, a common practice in construction management, to explore participants' perspectives on proportional punitive measures for late payment severity using various research methodologies. Ereiba et al. (2004) [62] conducted a literature review on FGD implementation in construction, highlighting its adaptability and depth for future research. FGDs were employed by Bagiu et al. (2020) [63] to gather structured data on issues faced by project managers in implementing current methodologies, best practices, and improvement ideas. Arewa et al. (2023) [61] used an FGD to investigate participants' perceptions of proportional punitive measures for late payments in construction. Seven methods are available for conducting FGDs [58]. Small and online focus groups were used in the study, and this allowed for the development of small, highly experienced groups [58].

The Delphi technique is a systematic approach and entails distributing questionnaires to the participants, who are subject matter experts, in several iterative rounds. Feedback is given to participants after each round via statistical analysis of group responses to offer a consensus stance [64]. Since the replies are anonymous, individuals can answer honestly without worrying about what their friends would think. Two or three rounds are often sufficient to build a consensus in Delphi studies [65]. Building on classical Delphi techniques, many different variations of Delphi techniques have been created [66–70]. It has been suggested that choosing the Delphi design depends on the circumstances dictated by the research topic [67,69]. The traditional Delphi technique consists of four phases: choosing experts, conducting the first round of surveys, conducting the second round of surveys, and combining expert viewpoints to produce a consensus. However, until a

final consensus is reached, processes (3) and (4) are iteratively repeated [64,69–71]. The implementation of the Delphi procedure and analysis is summarized in Figure 3.

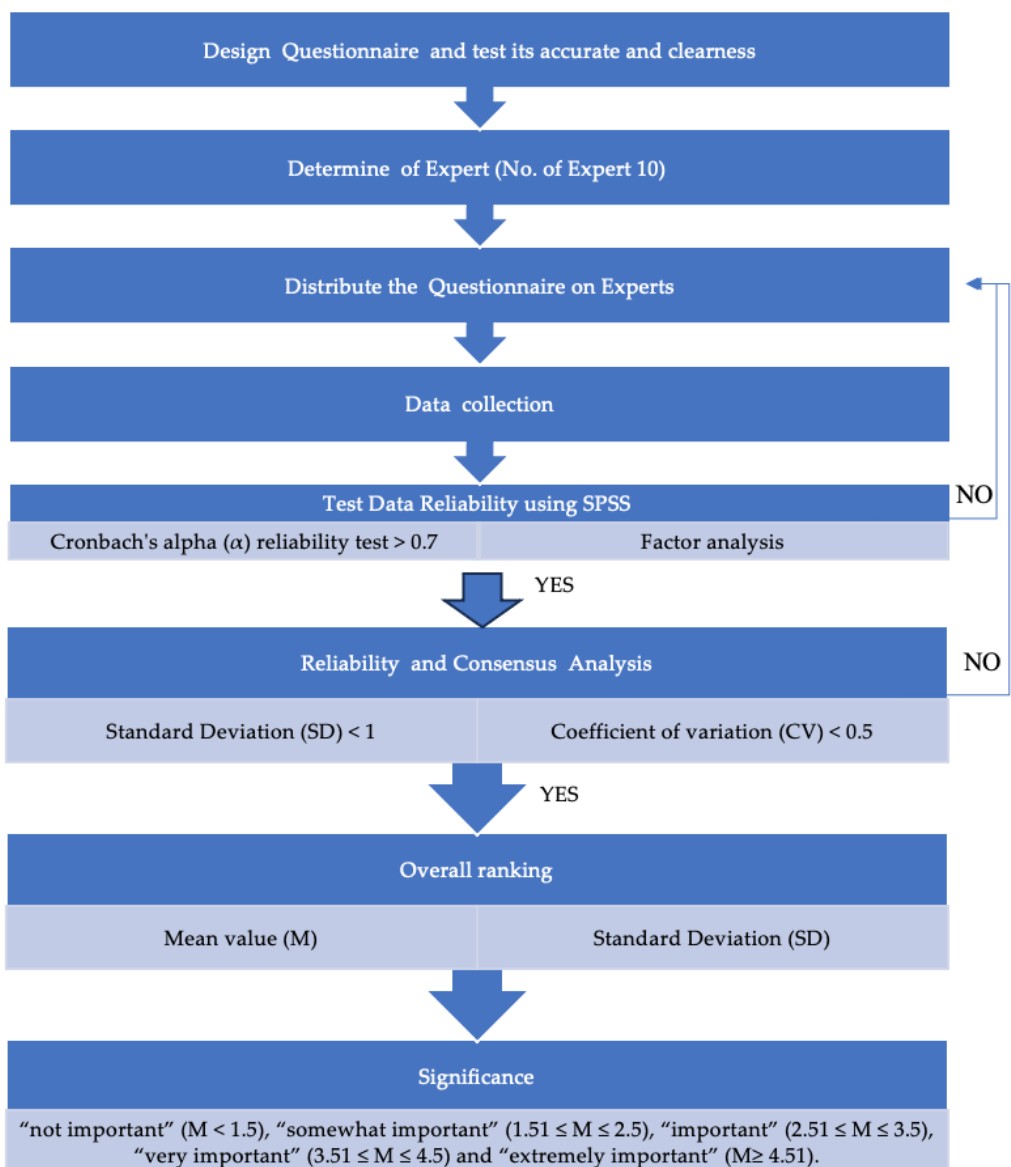

**Figure 3.** Delphi procedure and analysis chart.

## 4. Results

*4.1. The Development of the Lean Construction–Sustainable Development Goals (SDGs) Interaction Matrix*

The lean construction–SDGs interaction matrix was developed using Table 2, which analyzes the literature. Note that the line below the sentence shows the sustainability aspect targeted within the study to make the connection of the lean construction principle to the sustainability element.

**Table 2.** Applying lean construction principles to achieve SDGs.

| | References | The Synergy between Lean Construction and Sustainability | Lean Principle | Impacts on SDGs and Their Targets |
|---|---|---|---|---|
| 1 | [45] | Using VSM to reduce or eliminate waste. | L1, L11 | 8.4; 12.2; 12.4; 12.5 |
| 2 | [39] | There are potential applications of the LPS for safety management. | L1, L2 | 3.4; 12.8; 17.16 |
| 3 | [21] | When companies use lean construction, the 5S framework is useful for enhancing workspaces in industrial and service operations. | L3, L9 | 9.2 |
| 4 | [72] | Mainstreaming off-grid solar energy policies and giving it sector investment priority can lead to human development and well-being, the construction of physical and social infrastructure, and the sustainable management of environmental resources. | L2 | 1.1; 7.1; 17.5; 17.7; 17.14 |
| 5 | [73] | Using lean construction tools (VSM), material flows were drawn, and the non-value-adding steps were determined. | L1, L2, L6, L8, L9 | 4.7; 7.3; 8.2; 8.4; 9.2; 9.4; 9.5; 11.6; 12.2; 12.8; 17.7 |
| 6 | [74] | The advantage of combining lean construction and sustainability is waste reduction. | L2 | 8.2; 8.4; 11.6; 12.2; 12.5; 12.6 |
| 7 | [75] | There are interactions between lean construction and green buildings in increased resource utilization, reduced waste, a reduction in carbon gas emissions, energy savings, and better time–cost performance. | L1 | 7.3; 8.4; 9.2; 11.6; 12.2; 12.4; 12.5 |
| 8 | [76] | The methodology combined lean construction principles with sustainable construction principles for reducing waste and increasing productivity. | L1, L2 | 7.3; 8.4; 11.6; 12.4; 12.5; 12.8 |
| 9 | [77] | Applying lean construction and sustainable principles can immediately help companies complete their tasks more quickly and with higher quality. | L1 | 8.4; 11.6 |
| 10 | [78] | Lean construction helps cut back on material waste and work hours, improving the situation and reducing or eliminating struck-by accidents. | L1, L4 | 9.2; 17.1 |
| 11 | [79] | BIM use helps pull planning, commitment tracking, integrated project delivery (IPD), and reduction in materials and working hours. | L4, L1 | 9.2; 12.2 |
| 12 | [80] | Value stream mapping (VSM) is adapted to reveal improvement opportunities that typically remain hidden, reducing waste by synchronizing production to the customer's needs. | L2, L9, L11 | 9.4; 9.5; 12.5; 12.6 |
| 13 | [22] | Off-site construction (OSC) strategies support the lean construction flow process, reduce completion times, and simplify the process. | L4, L5 | 12.2; 9.2 |
| 14 | [81] | The lean principles promote sustainable construction practices. | L1 | 8.4; 12.2; 12.4; 12.5 |
| 15 | [19] | There are interactions between lean construction and green buildings due to enhanced resource usage, reduced time costs, energy savings, and a reduction in greenhouse gas emissions. | L1, L4 | 8.4; 9.2; 11.6; 12.2; 12.4; 12.5 |
| 16 | [82] | A conceptual framework for lean construction and green buildings was created using BIM to add value, save money and time, and detect collisions. | L7, L8, L9, L10 | 4.7; 8.2; 9.2; 9.4; 9.5; 12.2; 12.8; 17.7 |
| 17 | [83] | Applied a prefabrication just-in-time (prefab-JIT) system to improve the process and increase its efficiency, reducing the environmental impact, including global warming, acidification, eutrophication, and smog formation. | L1, L10, L2, L4, L6 | 8.2; 9.2; 11.6; 12.4; 12.5 |
| 18 | [84] | Displayed a variety of connections between lean construction principles and green construction, such as the overall reduction of waste and the creation of value for the clients. | L1 | 11.6; 12.2; 12.5 |
| 19 | [85] | Waste reduction, value delivery, and cost savings are the main goals of high-performance lean and green building projects. | L1, L2 | 11.6; 12.5 |
| 20 | [86] | With lean construction principles, it is possible to build LEED-certified buildings that cost no additional expense. | L2, L5, L8, L9, L11, | 7.3; 11.6; 12.2; 12.5; 12.6; 12.8; 17.5; 17.7; 17.9; 17.14; 17.19 |
| 21 | [87] | Sustainability emphasizes resource minimization, system optimization, and wasteful building methods, whereas lean construction principles concentrate on waste reduction and maximization of value creation. | L1, L2, L5, L8 | 8.4; 9.2; 11.6; 12.2; 12.7; 12.8 |
| 22 | [88] | Lean construction and sustainability are compatible when it comes to removing non-value-adding activities that would otherwise have a negative influence on the environment. | L1 | 11.6; 12.4; 12.5 |
| 23 | [89] | Lean construction helps to reduce variabilities and improve workflow, continuous improvement, visualization, communication, the participation of employees in the processes, develop customer strategies, improve material flow, control defects, and improve quality and safety. | L1, L2, L3, L7, L9, | 11.6; 12.2; 12.5; 4.7; 8.2; 8.4; 9.2; 9.4 |
| 24 | [90] | Utilizing lean construction with sustainable buildings would save on waste by specifying functions, capabilities, and requirements in advance and can reduce CO$_2$ | L1, L2, L3, L4, L6 | 8.2; 8.4; 9.2;11.6; 12.2; 12.5 |
| 25 | [91] | JIT production allows us to minimize buffers and reduce the various sources of extra inventory. | L6 | 12.2 |
| 26 | [92] | They found the effect of the reduction of construction waste production on cost, where lean saving represents 14% of green costs. | L1, L2, L11 | 12.4; 12.5; 12.8; 17.5 |
| 27 | [14] | Higher savings and an improvement in health and safety conditions would arise from lean construction use. | L1, L2 | 3.4; 8.4; 8.8; 9.4 |
| 28 | [93] | Lean construction enables initial cost reduction, waste elimination, and operating efficiency improvements. | L1, L9 | 8.2; 8.4; 11.6; 12.5 |
| 29 | [94] | Construction and maintenance of green facilities are impacted using lean construction tools and procedures. | L1, L2 | 8.2; 11.6 |
| 30 | [95] | Process mapping affects transparency and the delivery of sustainable projects. | L7 | 11.6; 12.2; 12.8 |
| 31 | [96] | Using lean construction will reduce waste and increase efficiency. | L1, L2, L4, L8 | 9.2; 11.6; 12.5 |
| 32 | [97] | A "lean" benchmark can be developed to provide comparative measurements of CO$_2$ for precast concrete products. | L11 | 9.5; 12.6 |
| 33 | [98] | Lean construction principles can be used to enhance productivity, safety, and quality. | L1, L2 | 8.2; 6.3; |
| 34 | [99] | Prefabrication can help standardization, minimize time, enhance quality assurance, and cut down material waste. | L4, L1, L8, L2 | 9.2; 11.6; 12.2; 12.4; 12.5; |
| 35 | [43] | The Last Planner System (LPS) helps to lower plan variations. | L3, L4, L5, L6, L7, L8 | 8.4; 9.2; 12.2; 12.5 |
| 36 | [42] | There is a waste reduction impact of prefabrication. | L2, L9 | 1.1; 12.5; 12.8 |

**Table 2.** *Cont.*

| | References | The Synergy between Lean Construction and Sustainability | Lean Principle | Impacts on SDGs and Their Targets |
|---|---|---|---|---|
| 37 | [100] | The efficacy of the suggested green lean approach, or value stream mapping (VSM), pertains to maximizing optimum resource use, cutting costs, improving quality, and reducing environmental effects. | L1, L11 | 8.4; 9.2; 12.2; 12.5; 12.4 |
| 38 | [101] | VSM helps to improve the environmental effects of construction operations. | L1, L10, L11 | 9.2; 12.2; 12.5; 8.4; 12.8; 17.9 |
| 39 | [102] | Improved time efficiency and process standardization. | L2 | 11.6; 12.5; 17.17 |

An interaction matrix in Supplementary Materials S1 was created by classifying 39 studies on lean construction principles and SDGs. The categories aligned with Barbier and Burgess's classification [103], finding 21 social, 29 economic, and 38 environmental SDG targets.

### 4.2. Focus Group Discussion Analysis Results to Validate the Interaction Matrix

The literature review analysis showed that there are 88 interactions between lean construction principles and SDGs. The studies tend to concentrate mostly on construction projects or activities, potentially leaving out some interactions. To fill this gap, a focus group discussion (FGD) session was held to identify novel interactions and confirm the findings of the first matrix. Table 3 includes the descriptive statistics related to experts. Experts were chosen because of their experience and knowledge about lean construction and SDGs.

**Table 3.** Respondent profiles (FGD).

| Expert ID | Profession | Education | Experience (Year) |
|---|---|---|---|
| EX1 | Academic | Architect, PhD Degree | CI: >20, LCp:10, SDG: 4 |
| EX2 | Academic | Civil Engineer, PhD Degree | CI: >20, LCp:14, SDG: 7 |
| EX3 | Academic | Civil Engineer, PhD Degree | CI: >20, LCp:15, SDG: 6 |
| EX4 | Project Manager | Architect, MSc. Degree | CI: >15, LCp:10, SDG: 4 |
| EX5 | Project Director | Civil Engineer, PhD Degree | CI: >20, LCp:16, SDG: 6 |

CI: construction industry; LCp: lean construction principles; SDG: Sustainable Development Goal.

In the literature review analysis, a total of 84 interactions between lean construction and Sustainable Development Goals (SDGs) were detected. Through FGD, an additional 375 interactions were unveiled (previously unrecognized relationships between lean construction and SDGs). They are denoted in Table 4 as colored. In total, this study documented 459 interactions (encompassing economic (188), social (71), and environmental (11) aspects) between SDGs and lean construction principles because of FGD. However, to evaluate the impacts of these interactions, the Delphi Survey was employed like shown in Supplementary Materials S2.

**Table 4.** Lean construction–SDG interaction matrix.

| SDG | Target | L1 | L2 | L3 | L4 | L5 | L6 | L7 | L8 | L9 | L10 | L11 |
|---|---|---|---|---|---|---|---|---|---|---|---|---|
| | | SDG1 End poverty in all its forms everywhere | | | | | | | | | | |
| 1 | 1.1 | | ■ | | ■ | | | | | ■ | ■ | |
| | 1.2 | | ■ | | | | | | ■ | | ■ | ■ |
| | 1.3 | | | | | | | | | | | |
| | 1.4 | | | | | | | | ■ | ■ | | |
| | | SDG2 End hunger, achieve food security, and improve nutrition | | | | | | | | | | |
| 2 | 2.1 | ■ | | | ■ | | | ■ | | | | |
| | 2.3 | ■ | | | ■ | | | ■ | | | | |
| | 2.4 | | | | | | | | | | | |
| | 2.5 | | ■ | | | | | | | | | |
| | | SDG3 Ensure healthy lives and promote well-being for all at all ages | | | | | | | | | | |
| 3 | 3.4 | ■ | ■ | | | | | ■ | | ■ | | ■ |
| | 3.9 | ■ | | | | | | | | ■ | | ■ |
| | | SDG4 Ensure inclusive and equitable quality education and promotes learning opportunities | | | | | | | | | | |
| 4 | 4.5 | ■ | | | | | | | | ■ | | |
| | 4.7 | ■ | | | | | | | | ■ | | |
| | | SDG5 Achieve gender equality and empower all women and girls | | | | | | | | | | |
| 5 | 5.1 | | ■ | | | | | | | ■ | ■ | |
| | 5.2 | | ■ | | | | | | | ■ | | |
| | 5.3 | | | | ■ | | | | | | | |
| | 5.4 | | ■ | | | | | | | ■ | | |
| | 5.5 | | | | | ■ | | | | ■ | | |
| | 5.6 | | ■ | | | | | | | ■ | ■ | |

**Table 4.** *Cont.*

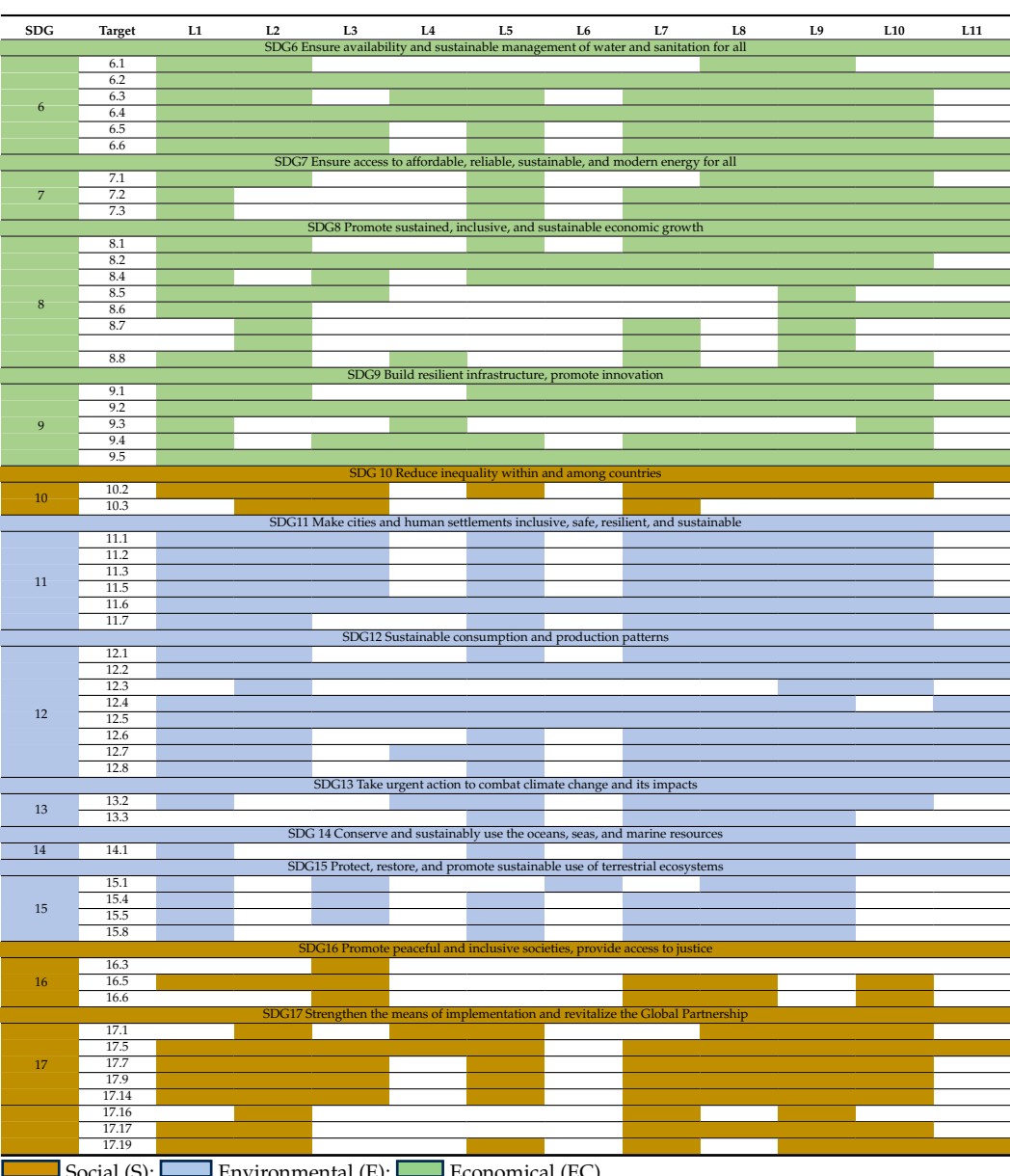

Social (S); Environmental (E); Economical (EC).

## 4.3. The Delphi Analysis

### 4.3.1. The Use of the Two-Round Delphi Technique

The Delphi technique, like the classic version, requires iterative rounds to build a consensus. The questionnaire used in the Delphi technique was created according to the FGD analysis results. In the literature, some researchers suggested the use of the first Delphi technique round to uncover new interactions. However, FGDs offer more reliable results for uncovering unknown phenomena [64,104]. Moreover, Hsu and Lin (2013) [64] and Ameyaw et al. (2016) [104] stated that the first round of the Delphi technique can be replaced with FGDs. For lean construction–SDG synergy assessment via Delphi, a purposive sampling technique was used in the selection of the Delphi experts [105]. Delphi panel size varies and depends on factors like theoretical saturation, homogeneity, problem scope, and research team capacity. So, the sample size and its randomization are irrelevant when using the Delphi technique [106]. Typically, 7 to 20 experts can participate in over 2–3 rounds to build a consensus [104,107–111].

While selecting experts for Delphi analysis, these selection criteria were used: (i) respondents with significant experience and leadership in the construction industry;

(ii) respondents who have applied straightforward, sustainable practices in current or previous construction projects; and (iii) respondents who have a firm grasp of SDG concepts. Ten participants were selected from fifteen based on criteria.

The Delphi survey spanned four months. In the first round of Delphi analysis, the interactions/synergies between SDGs and lean construction were evaluated by using a 5-point Likert scale (1 for "strongly disagree" and 5 for "strongly agree"). After analysis of the collected data, the follow-up second round was held to build a consensus. Before collecting data in the second round, the experts were also informed of the preliminary findings of the first round, allowing for modifications to the ratings in the next round. The experts' confidentiality was always maintained. Analytical scrutiny was then applied to the collated expert responses.

### 4.3.2. The Descriptive Statistics of Experts

The experts consisted of participants who have experience in lean construction and sustainability practices in prior or ongoing projects. Ten experts from five different nations were chosen. Ten experts comprising five academicians and five practitioners, including engineers and project managers, participated in the panel to obtain comprehensive insights into the topic. Table 5 contains descriptive statistics for the experts.

**Table 5.** Descriptive statistics of experts.

| Country | No. | Expert Experience | No. |
|---|---|---|---|
| USA | 1 | 5–10 y | 1 |
| Turkey | 4 | 10–20 y | 3 |
| UEA | 2 | >20 | 6 |
| Egypt | 1 | | |
| Iraq | 2 | | |
| **The size of the company you work for** | | **Lean construction experience** | No. |
| Small | 1 | 1–5 y | 2 |
| Medium | 4 | 6–10 y | 3 |
| Large | 5 | 11–20 y | 5 |
| **Position** | | **SDGs experience** | No. |
| Academic | 5 | 1–5 y | 7 |
| General Manager | 1 | >5 | 3 |
| Project Manager | 1 | | |
| Business Development Engineer | 2 | | |
| Project Director | 1 | | |
| **International experience of the company** | | **Background** | No. |
| 1–5 y | 1 | Civil engineer | 7 |
| 5–10 y | - | Architect | 3 |
| 10–20 y | 4 | | |
| >20 | 5 | | |
| **Institution Of Expert** | No. | **Education status** | No. |
| Private sector | 3 | Bach. | 1 |
| University | 5 | MSc. | 2 |
| Other Public Institutions | 2 | PHD | 7 |

### 4.3.3. Data Analysis

The responses of the experts were analyzed using descriptive and inferential statistical methods, i.e., factor analysis for reliability tests, overall ranking, significance; the Cronbach's alpha reliability test; coefficient of variation (CV); standard deviation (SD); and mean scores (M) values. Afterward, the results were discussed.

- Data reliability analysis and consensus

The Cronbach's alpha ($\alpha$) reliability test, as shown in Equation (1), helps to determine whether the questionnaire and associated scale measure the correct structure and to check internal consistency. Its value spans from 0 to 1. A score of 0.7 or higher is considered appropriate. $\alpha > 0.80$ indicates good internal consistency [112,113]. The alpha ($\alpha$) values for Delphi's first

and second rounds were 0.96 and 0.974. IBM SPSS 28 has been used as shown in the reliability test for second-round and factor analysis, given in Supplementary Material S3 and S4.

$$\alpha = K/(K-1)\ [1-(\Sigma\sigma^2/\sigma^2\ \text{total})], \tag{1}$$

where $\Sigma\sigma^2 k$ is the sum of the variances of all the items (k), and $\sigma^2$ is the variances.

In addition to the internal consistency, the study relies on advanced statistical methods using factor analysis to show the results, as shown in Supplementary Material S4, where the matrix of factors contained all the variables (1–10) under study, which are considered acceptable as their value was more significant than 5%. This is one of the indications of the effectiveness of the data obtained from the statistics from the respondents' answers, which is an important measure that the researcher can rely on.

- Consensus levels

Quantitative outcomes from the Delphi technique can be evaluated in terms of consensus levels. Common analytical tools, like mean values and standard deviation (SD), are employed to gauge the significance of attributes and sub-attributes [114].

Research indicates that a 5-point Likert scale with SD < 1 is indicative of minimal rating dispersion [115]. SD was lower than 1 for most interactions, except 19 interactions "1.2L10, 5.1L7, 6.6L2, 9.1L2, 9.2L8, 9.2L, 11, 9.3L10, 10.2L2, 10.2L8, 11.6L11, 12.2L11, 12.3L9, 12.4L2, 12.5L2, 12.7L11, 12.8L11, 15.5L5, 15.5L7, 17.14L2", that are associated with multiple goals and targets.

The acceptance analysis in the first round showed a consensus or near-consensus on most of the interactions. However, it was observed that some standard deviation values were larger than 1. This required us to carrying out the second round of the Delphi survey.

In addition, the coefficient of variation (CV) was tested (Equation (2)) [116], which can help compare the overall accuracy of the data [116]. It is the procedure by which to obtain reliability in a Delphi study [117] where:

$$CV = \{\text{Stander deviation/mean}\}\ <\ 0.5. \tag{2}$$

Most CV values fell below the threshold of 0.5, except for a few interactions involving multiple goals and targets, including "1.2L10, 5.1L7, 6.6L2, 9.1L2, 9.2L8, 9.2L, 11, 9.3L10, 10.2L2, 10.2L11, 12.2L11, 12.3L9, 12.4L2, 12.5L2, 12.7L11, 12.8L11, 15.5L5, 15.5L7, and 17.14L2".

The same panelists and the questionnaire from the first round were used in round two. To help with consensus-building—which is a key objective of the Delphi technique—researchers disclosed information on the opinions of other panelists. The panelists' ability to reach a consensus was aided by statistics that summarized the experts' earlier comments and the consensus from the first round. The second Delphi survey's findings show that the respondent groups agrees.

- Overall ranking of the SDGs

The mean value (M) and standard deviation (SD) were calculated to evaluate the level of importance of attributes and sub-attributes [114]. The mean value is used to determine the importance of interaction. The higher the mean value, the greater the acceptance of interaction from the perspective of the experts. The overall the M, SD, and CV of the interaction for the second round of the Delphi survey are shown in Supplementary Material S5 (due to the limited number of pages).

We used both their mean scores (M) and the standard deviation (SD) values in ranking the 459 goals based on the responses from the expert panel across the two rounds of the Delphi survey. The SD value is considered when two or more factors have the same value for their mean score. Therefore, the factor with the smaller SD value is assigned a higher rank; otherwise, if the SD is the same, the elements will maintain the same rank [118]. In the first-round Delphi survey, the mean score for the 459 ranked interactions was M = 5 (SD = 0) for "6.2L2, 6.2L8, 8.2L1, 12.7L8, 12.7L9" to M = 1 (SD = 0) for "1.3L3, 1.4L3, 2.1L4,

2.3L4, 2.3L5, 2.4L4, 2.4L6, 2.5L2, 2.5L5, 2.5L6, 3.4L2, 3.9L11, 4.5L3, 4.7L2, 4.7L3, 4.7L7, 4.7L9, 5.2L1, 5.2L3, 5.3L1, 5.3L4, 5.3L9, 5.4L1, 5.4L9, 5.6L1, 5.6L3, 8.5L3, 8.7L2, 12.3L2, 12.3L10".

In the second-round Delphi survey, the mean score for the 459 ranked interactions ranged from M = 5 (SD = 0) for the same infraction in the first round to M = 1.1 (SD = 0.32) for 2.1L5, 2.3L6, 2.4L7, 2.5L7, 5.4L3, 1.3L3, 1.4L3, 2.1L4, 2.4L4, 2.5L2, 3.4L2, 4.5L3, 4.7L3, 4.7L9, 5.2L1, 5.3L1, and 5.4L1. The Delphi technique's core aim is to achieve consensus among respondents after the closure of the survey rounds.

- Significance of the SDGs

The data used for analysis are based on the 459 identified interactions' mean score values from the second round of the Delphi survey. Furthermore, these interactions were ranked using scale interval interpretation [119] as follows: "not important" (M < 1.5); "somewhat important" (1.51 ≤ M ≤ 2.5); "important" (2.51 ≤ M ≤ 3.5); "very important" (3.51 ≤ M ≤ 4.5); and "extremely important" (M ≥ 4.51) [23,118]. Table 6 shows the most significant interactions from the second-round Delphi survey according to interaction type (economic, environmental, and social) in green, blue, and brown, respectively. A summary of the interaction significance is provided in Supplementary Material S5 in descending order.

**Table 6.** The extremely important interactions for the second round of the Delphi survey.

| | Interaction | M | SD | CV | Rank | Significance |
|---|---|---|---|---|---|---|
| 1 | 6.2L2 | 5 | 0 | 0 | 1 | Ext.Imp. |
| 2 | 6.2L8 | 5 | 0 | 0 | 1 | Ext.Imp. |
| 3 | 8.2L1 | 5 | 0 | 0 | 1 | Ext.Imp. |
| 4 | 12.7L8 | 5 | 0 | 0 | 1 | Ext.Imp. |
| 5 | 12.7L9 | 5 | 0 | 0 | 1 | Ext.Imp. |
| 6 | 8.2L4 | 4.9 | 0.32 | 0.065 | 2 | Ext.Imp. |
| 7 | 8.2L8 | 4.9 | 0.32 | 0.065 | 2 | Ext.Imp. |
| 8 | 8.2L9 | 4.9 | 0.32 | 0.065 | 2 | Ext.Imp. |
| 9 | 8.2L10 | 4.9 | 0.32 | 0.065 | 2 | Ext.Imp. |
| 10 | 12.2L1 | 4.9 | 0.32 | 0.065 | 2 | Ext.Imp. |
| 11 | 12.5L10 | 4.9 | 0.32 | 0.065 | 2 | Ext.Imp. |
| 12 | 12.8L5 | 4.9 | 0.32 | 0.065 | 2 | Ext.Imp. |
| 13 | 12.8L8 | 4.9 | 0.32 | 0.065 | 2 | Ext.Imp. |
| 14 | 12.8L9 | 4.9 | 0.32 | 0.065 | 2 | Ext.Imp. |
| 15 | 11.1L2 | 4.8 | 0.42 | 0.088 | 3 | Ext.Imp. |
| 16 | 11.7L8 | 4.8 | 0.42 | 0.088 | 3 | Ext.Imp. |
| 17 | 12.1L8 | 4.8 | 0.42 | 0.088 | 3 | Ext.Imp. |
| 18 | 12.1L9 | 4.8 | 0.42 | 0.088 | 3 | Ext.Imp. |
| 19 | 12.2L5 | 4.8 | 0.42 | 0.088 | 3 | Ext.Imp. |
| 20 | 12.2L8 | 4.8 | 0.42 | 0.088 | 3 | Ext.Imp. |
| 21 | 12.2L9 | 4.8 | 0.42 | 0.088 | 3 | Ext.Imp. |
| 22 | 12.7L5 | 4.8 | 0.42 | 0.088 | 3 | Ext.Imp. |
| 23 | 12.7L7 | 4.8 | 0.42 | 0.088 | 3 | Ext.Imp. |
| 24 | 12.8L7 | 4.8 | 0.42 | 0.088 | 3 | Ext.Imp. |
| 25 | 16.5L7 | 4.8 | 0.42 | 0.088 | 3 | Ext.Imp. |
| 26 | 7.3L9 | 4.8 | 0.42 | 0.088 | 3 | Ext.Imp. |
| 27 | 11.3L9 | 4.8 | 0.42 | 0.088 | 3 | Ext.Imp. |
| 28 | 11.5L7 | 4.8 | 0.42 | 0.088 | 3 | Ext.Imp. |
| 29 | 11.6L5 | 4.7 | 0.48 | 0.103 | 4 | Ext.Imp. |
| 30 | 11.6L9 | 4.7 | 0.48 | 0.103 | 4 | Ext.Imp. |
| 31 | 12.1L5 | 4.7 | 0.48 | 0.103 | 4 | Ext.Imp. |
| 32 | 12.5L9 | 4.7 | 0.48 | 0.103 | 4 | Ext.Imp. |
| 33 | 15.4L5 | 4.7 | 0.48 | 0.103 | 4 | Ext.Imp. |
| 34 | 11.6L8 | 4.7 | 0.48 | 0.103 | 4 | Ext.Imp. |
| 35 | 12.2L7 | 4.7 | 0.48 | 0.103 | 4 | Ext.Imp. |
| 36 | 12.4L8 | 4.7 | 0.48 | 0.103 | 4 | Ext.Imp. |
| 37 | 15.1L8 | 4.7 | 0.48 | 0.103 | 4 | Ext.Imp. |
| 38 | 15.4L1 | 4.7 | 0.48 | 0.103 | 4 | Ext.Imp. |
| 39 | 17.19L9 | 4.7 | 0.48 | 0.103 | 4 | Ext.Imp. |
| 40 | 6.4L9 | 4.6 | 0.52 | 0.112 | 5 | Ext.Imp. |
| 41 | 6.5L8 | 4.6 | 0.52 | 0.112 | 5 | Ext.Imp. |
| 42 | 9.4L9 | 4.6 | 0.52 | 0.112 | 5 | Ext.Imp. |
| 43 | 11.7L9 | 4.6 | 0.52 | 0.112 | 5 | Ext.Imp. |
| 44 | 15.8L5 | 4.6 | 0.52 | 0.112 | 5 | Ext.Imp. |
| 45 | 11.6L1 | 4.6 | 0.52 | 0.112 | 5 | Ext.Imp. |
| 46 | 12.5L1 | 4.6 | 0.52 | 0.112 | 5 | Ext.Imp. |

Ext.Imp. = extremely important. ☐ Social (S); ☐ Environmental (E); ☐ Economical (EC).

## 5. Analysis of Results

The findings of this study highlight that there are many interactions between SDGs and lean construction. The table given in Supplementary Material S5 offers a comprehensive visualization of the ramifications of applying lean construction principles in SDG attainment, encompassing both "extremely important" and "very important" interlinkages. The analysis further unveils 63 "extremely important" interactions, indicated by rankings ranging from 1 to 6 from 83 ranks. Figure 4 conveys a visual representation of the influence of lean construction principles on the accomplishment of ten out of the seventeen SDGs as "extremely important" relations. These findings indicate that the extremely important interactions between lean construction principles and the SDGs cluster around SDGs 6, 8, 11, 12, and 15. However, lesser interactions were observed within SDG 7, 9, 14, 16, and 17. The supplementary table in Supplementary Material S5 ranks this relationship.

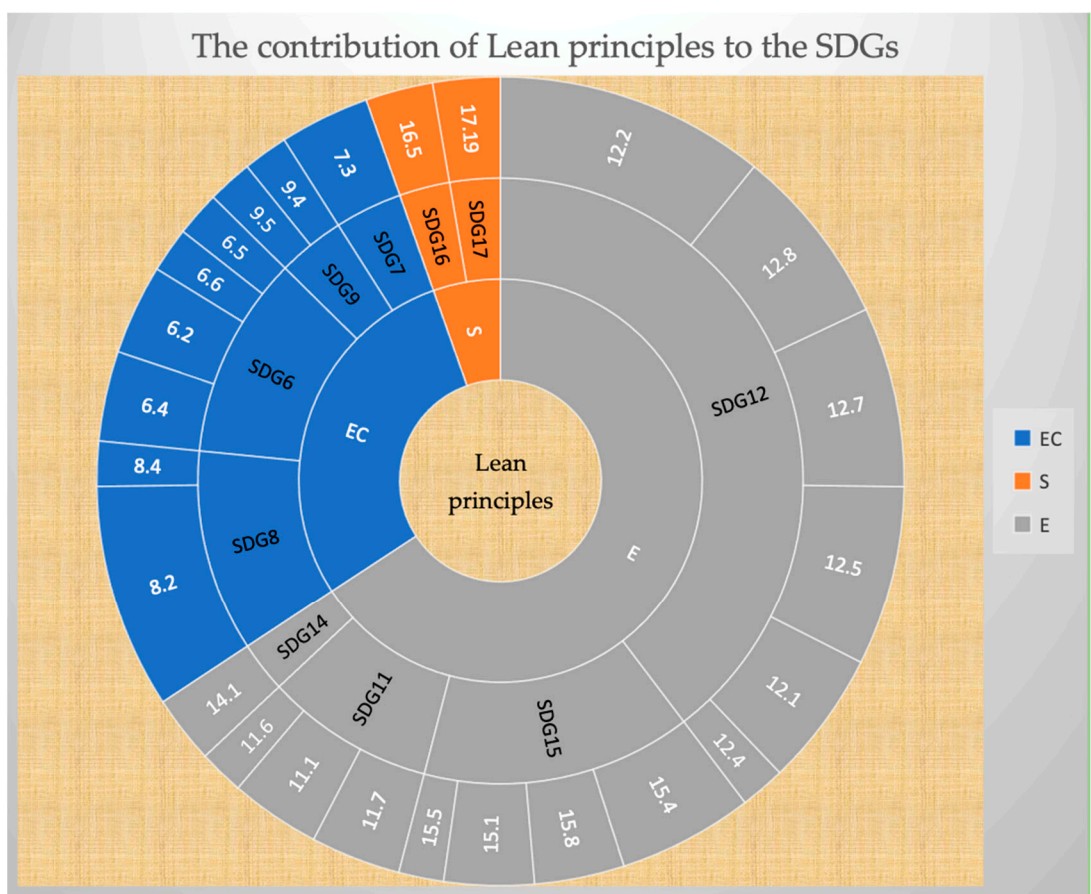

**Figure 4.** The extremely important contribution of lean principles to the SDGs.

The highest interaction levels with lean construction principles are found in SDG 12, "Responsible Consumption and Production", indicating lean construction's potential to influence sustainable practices.

According to the Delphi analysis, the most important SDG is SDG 6, "Ensure availability and sustainable management of water and sanitation for all", followed by SDG 8, "Promote sustainable economic growth", and SDG 12 "Responsible Consumption and Production". Figure 5 depicts "extremely important" interactions between the lean construction principle and SDGs.

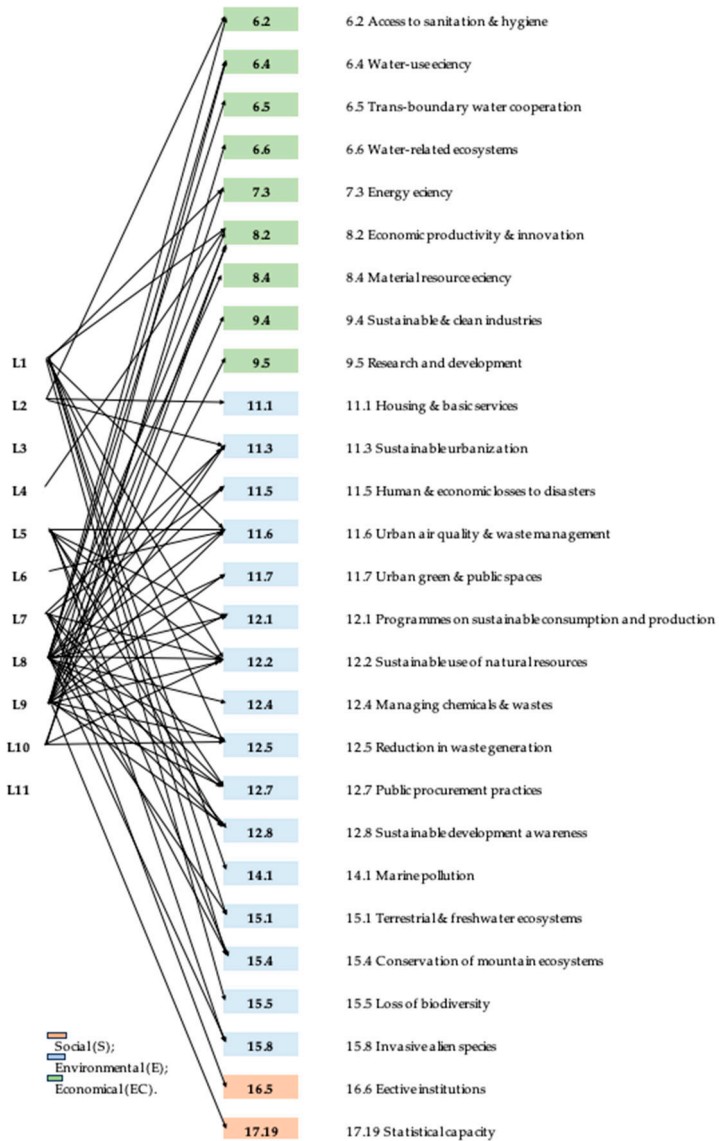

**Figure 5.** The extremely important linkages of lean principles with the SDGs.

Specific to lean construction principles, the most common impact is intensified around principles L1 (9 interactions), L9 (14 interactions), and L8 (17 interactions), followed by L7, L2, L4, and L6, as shown in Figure 5. Also, Supplementary Material S5 shows that principles L3 and L11 have a "very important" impact on SDGs, with 17 links for principle L3 and 1 for principle L11. Furthermore, analysis results showed that by leveraging lean construction principles L1, L9, and L8, a remarkable 63% of "extremely important" relations can be reached. Furthermore, analysis results showed that by leveraging lean principles L1, L9, and L8, (reducing non-value-adding activities, focusing on all processes, and integrating continuous improvement into processes) a remarkable 63% of "extremely important interactions" can be effectively attained. In other words, a substantial portion (30%) of all interactions classified as extremely important and very important can be achieved by strategically applying lean construction principles L1, L9, and L8. This strategic approach acknowledges the principles' priority, as deduced from their rankings.

This research reveals through 63 "extremely important" interactions a strong emphasis on the environmental dimension (45 interactions), indicating a prominent focus on ecological considerations and mitigating negative repercussions. Economic interactions come in second with sixteen interactions, while social interactions account for two. This distribution emphasizes the environmental sensitivity inherent in lean construction methods.

Supplementary Material S5 gives further insights according to 215 "very important" interactions, which span from rank 7 to 33. The economic aspect of the SDGs has the highest concentration of contacts (108), followed by environmental issues (84 interactions) and the social sphere (59 interactions). This design emphasizes the interdependence of lean construction principles and economic imperatives while addressing environmental and social problems.

Overall, the dominance of economic and environmental issues is consistent with the global trend towards sustainable environmental practices and contributes significantly to economic growth and efficiency. As an integrated framework, lean construction principles assist the SDGs' diverse objectives. Table 6 summarizes these connections, and Table 7 summarizes the number of interactions and their classification.

**Table 7.** A summary of the number and classification of interactions.

| Importance Level | Total No. | Sequence | Rank Range | E | EC | S |
|---|---|---|---|---|---|---|
| Extremely important | 63 | 1–63 | 1–6 | 45 | 16 | 2 |
| Very important | 251 | 64–314 | 7–33 | 84 | 108 | 59 |
| Important | 81 | 315–395 | 34–63 | 22 | 39 | 20 |
| Somewhat important | 20 | 396–415 | 64–79 | 3 | 2 | 15 |
| Not important relation | 44 | 416–459 | 80–83 | 2 | 8 | 34 |

S: social; E: environmental; EC: economical.

Figure 5 visually reinforces the alignment with environmental features, while Figure 6 shows a summary of the classification of interactions. It demonstrates how ten of the seventeen SDGs—SDGs 6, 7, 8, 9, 11, 12, 14, 15, 16, and 17—have an extremely important link to lean construction principles.

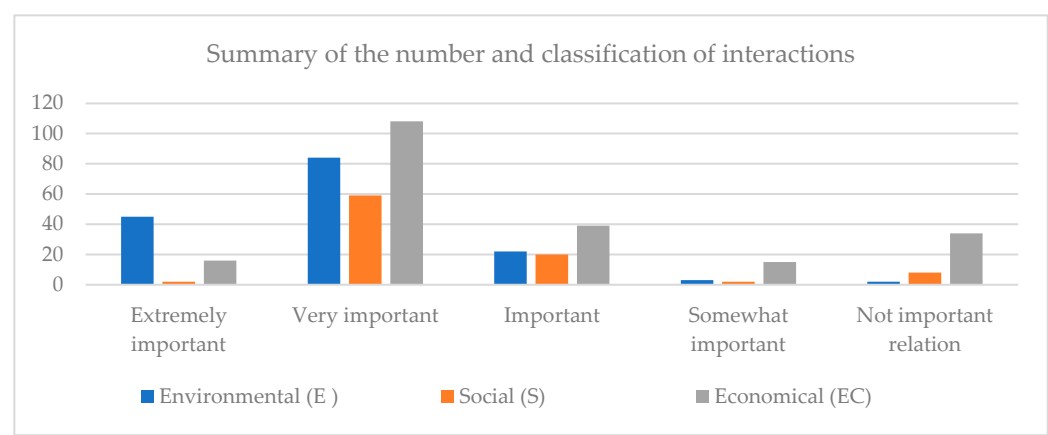

**Figure 6.** Summary of the classification of interaction.

Also, both "extremely important" and "very important" links in Figure 7 show the effect of using lean construction principles on achieving SDGs. These thorough results highlight the essential part played by lean construction principles in promoting the global sustainability agenda for 2030. Also, the strategic triad of lean construction principles—L1, L9, and L8—emerged as key in fostering "extremely important" interactions.

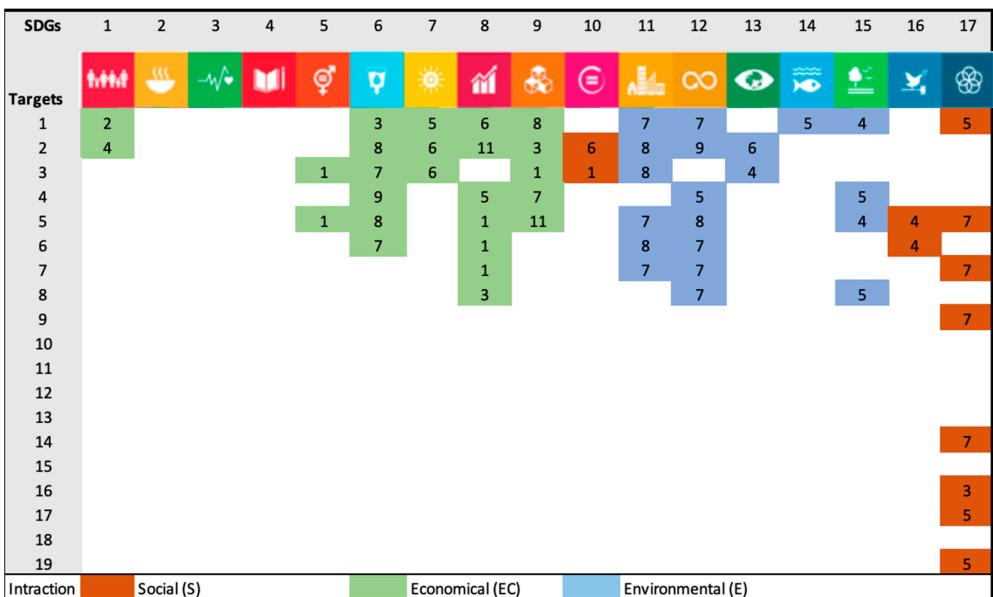

**Figure 7.** SDGs and targets affected by extremely important and very important interlinkages.

## 6. Discussion

These empirical findings directly address the research objectives by identifying and evaluating the key interactions between lean principles and the identified SDGs, thus clarifying the most influential principles in promoting SDGs in construction projects. For example, the results reveal that SDG 12, "Responsible Consumption and Production", shows the highest levels of engagement with lean principles. This indicates the potential of lean principles to influence sustainable practices, in line with the research objective of assessing the impact of lean principles on the SDGs.

The analysis shows that a significant portion of "critically important interactions" can be achieved through the strategic application of specific simple principles such as "minimizing non-added value, focusing on all processes, and continuous improvement". This underscores the importance of prioritizing certain principles to maximize their impact on sustainability goals.

The distribution of interactions emphasizes the environmental dimension of lean principles, with a strong focus on mitigating negative consequences. This emphasizes the ability of lean principles to contribute to environmental sustainability—as was proven by both Francis and Thomas (2020) [120] and Fuenzalida et al. (2016) [121]—and then achieve sustainable development goals in construction projects.

Focusing on the economic aspects of the SDGs highlights the interconnections between lean principles and economic imperatives, emphasizing their role in driving economic growth and efficiency while addressing environmental and social problems; this is what was proven by Halkos and Gkampoura (2021) [9].

The study results showed that there is extreme interaction between "Focus on all processes (L8)" and "promoting comprehensive sanitation approaches (SDG 6.2)". The lean construction methodology aims to utilize resources such as water, energy, raw materials, etc., efficiently by considering all processes and optimizing resource utilization. For example, sewerage systems, water treatment systems, etc. are directly connected to the building utilization systems. Depending on failures in these systems, the spread of disease can be unavoidable. Therefore, the study results agree that the balance between resource utilization or the reuse of water and the well-being of occupants should be carefully provided [90,93].

Also, employing process monitoring and control systems (L8) will lead to efficient resource utilization in sanitation projects, streamlining operations and minimizing costs [73]. Prioritizing customer needs (L2) in sanitation projects ensures efficient resource allocation,

reducing costs and promoting sustainability. By emphasizing value creation, lean principles drive innovation, as approved by Meshref et al. [35], and Aslam et al. [89], enabling affordable solutions for equitable sanitation access. This approach not only enhances project economics but also addresses the specific needs of vulnerable populations, aligning with SDG 6.2.

The strategic implementation of minimizing non-value-adding activities (L1) exhibits a profound economic connection with SDG Target 8.2, which emphasizes the imperative to achieve elevated economic productivity through diversification, technological upgrades, and innovation by efficiently curtailing time- and resource-intensive processes; this is what some research has suggested when matches with lean construction have contributed to heightened economic productivity [33,98].

Focusing on processes (L8) by implementing monitoring and control systems and embedding continuous improvement (L9) are important waste-reduction activities for the reduction of the construction industry's impacts on the environmental landscape [89,120]. These lean construction principles facilitate the reaching of SDG 12 (responsible consumption and production) in the construction industry.

Lean construction principles, particularly emphasizing the transparency of production processes (L7), contribute to achieving SDG 16.5 by creating an environment where bribery is minimized, and this is confirmed by a study by Klotz and Horman (2007) [95], which mentioned that documentation and reporting reduce opportunities for corruption and bribery, ensuring accessible information for all stakeholders [95]. Also, reducing diversity and uncertainty in processes (L3) establishes clear and standardized procedures, reducing corruption potential. Focusing on all processes (L8) ensures comprehensive oversight, making it difficult for illicit activities to go unnoticed. Simplifying processes (L5) minimizes opportunities for fraudulent practices. Analyzing and optimizing workflows (L10) identifies vulnerabilities, strengthening corruption prevention. Finally, when lean construction principles are applied, the selection of contractors and suppliers becomes more objective, fostering transparent, accountable, and corruption-resistant systems aligned with SDG 16.5.

Incorporating continuous improvement (L9) into construction processes also improves safety, health, and community comfort [98]. Moreover, continuous improvement requires robust performance measurement systems. SDG 17.19 also requires sustainable progress measures by 2030. Aligning L9 with SDG 17.19, the construction industry can enable a better assessment of its effect on sustainable practices.

This indicates the potential of lean principles to influence sustainable practices, in line with the research objective of assessing the impact of lean principles on the SDGs.

The distribution of interactions emphasizes the environmental dimension of lean principles, with a strong focus on mitigating negative consequences. This emphasizes the ability of lean principles to contribute to environmental sustainability and achieve Sustainable Development Goals in construction projects.

Focusing interactions on the economic aspects of the SDGs highlights the interconnections between lean principles and economic imperatives, emphasizing their role in driving economic growth and efficiency while addressing environmental and social problems.

## 7. Contributions of the Findings

This study aims to present a novel perspective, not extensively covered in prior research, concerning the direct and indirect contributions of buildings to the SDGs. Additionally, it delves into unexplored synergies between SDGs and lean construction principles. Consequently, researchers can uncover synergistic opportunities by considering their complementarity. Furthermore, leveraging the existing theory of lean construction principles can facilitate the adoption of SDGs given the widespread implementation and awareness of these principles in industries, thereby enhancing SDG-related efforts by academia and public/private institutions. Moreover, both concepts necessitate intensive collaboration among policymakers, industry practitioners, researchers, etc. These study findings can

serve as valuable insights for construction companies and policymakers seeking to mitigate the environmental impact of the construction industry.

Conducted specifically for the construction industry, this study acknowledges the high awareness of lean construction but the relatively low awareness of Sustainable Development Goals (SDGs). To tackle this issue, experienced academics and practitioners were engaged to explore the synergies between the two concepts. However, offering more examples of SDGs could enhance our understanding of the impact of lean construction on SDGs.

## 8. Conclusions

This research has established significant connections between the sustainable built environment (SBE), which uses lean construction principles, and specific Sustainable Development Goals (SDGs). This study employs qualitative and quantitative analysis using the literature review, FGD, and Delphi analysis. Moreover, statistical analysis methods, including Cronbach's alpha, mean, coefficient of variation, and score ranking, were used.

The research identified 63 "extremely important" interactions, particularly in SDGs 6, 7, 8, 9, 11, 12, 16, and 17. The strategic triad of lean construction principles, "Reducing non-value-adding, focusing on all processes, and continuous improvement", emerged as pivotal in fostering these essential interactions, particularly in promoting responsible consumption, production, and environmental protection across economic, environmental, and societal dimensions. Lean construction principles, encompassing customer needs and continuous improvement, align seamlessly with the SDGs promoting equitable sanitation access (SDG 6.2), economic productivity (SDG 8.2), and sustainable procurement (SDG 12.7). The efficiency and resource optimization driven by lean construction principles provide robust support for economic growth and diversification, while continuous improvement acts as a catalyst for fostering innovation and technological progress. The optimization of construction workflows not only enhances efficiency but also promotes sustainable production and resource utilization in alignment with SDG 8.2. The incorporation of continuous improvement aligns harmoniously with SDG 12.9, emphasizing the identification and implementation of sustainable enhancements in construction production. Lean construction principles play a pivotal role in advancing sustainable public procurement practices (SDG 12.7) by reducing waste and enhancing cost-effectiveness and resource efficiency. Furthermore, the emphasis on transparency in production processes, a key aspect of lean construction principles, significantly contributes to the attainment of SDG 16.5 by minimizing opportunities for corruption and bribery. The practice of clear documentation and reporting enhances overall accountability, thereby reducing information imbalances.

The study results enable the assessment of a building's contribution to SDGs, emphasizing the importance of lean construction principles in achieving these global objectives. This research introduces a novel perspective, an area not extensively explored in previous studies. The identified synergies between lean construction principles and SDGs are crucial for simultaneously advancing multiple goals, offering opportunities for collaborative projects and partnerships among stakeholders, and enhancing their SDG-related efforts. This research also contributes to the literature and expands our understanding of how lean principles in construction can directly or indirectly contribute to Agenda 2030, benefiting both academia and public/private institutions. As a further study, it will be useful to explore integrating BIM tools with lean principles to effectively reach SDGs.

**Supplementary Materials:** The following supporting information can be downloaded at: https://www.mdpi.com/article/10.3390/su16083502/s1, Supplementary Materials S1: Interaction Matrix Development—Literature reviews; Supplementary Materials S2: Delphi Survey; Supplementary Materials S3: Reliability test for 2ed round using SPSS; Supplementary Materials S4: Factor analysis using SPSS; Supplementary Materials S5: Second round of the Delphi survey. References [14,19,21,22,39,42,43,45,46,72–83,85–96,98–102] are cited in Supplementary Materials.

**Author Contributions:** S.H.: Conceptualization, Methodology, Formal analysis. Investigation, Writing—original draft, Writing—Review and Editing, Visualization; Z.I.: Conceptualization, Methodology, Validation, Resources, Writing—Review and Editing, Supervision; G.D.: Conceptualization, Methodology, Validation, Resources, Data curation, Writing—review and editing. All authors have read and agreed to the published version of the manuscript.

**Funding:** This research did not receive any specific grant from funding agencies in the public, commercial, or not-for-profit sectors.

**Institutional Review Board Statement:** Not applicable.

**Informed Consent Statement:** Not applicable.

**Data Availability Statement:** Data and any other data are contained within the article.

**Conflicts of Interest:** The authors report no conflicts of interest. The authors alone are responsible for the content and writing of this article.

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
