# Peer review of "Evaluating the Contribution of Lean Construction to Achieving Sustainable Development Goals"

_sustainability, doi:10.3390/su16083502_

Round 1

Reviewer 1 Report

Comments and Suggestions for Authors

Dear authors and editors. I can make the following minor comments on the text of the manuscript of the scientific article.

1. I recommend not using the abbreviation Lean Construction (LC), as it creates confusion with another well-known and more commonly used abbreviation Land Cover (LC).

2. Line 1. Have you correctly identified the type of publication. I think it's more of a review than an article.

3. The manuscript of a scientific article is not designed according to the rules of a scientific journal.

Author Response

We thank our valuable editor and all valuable reviewers for their constructive comments and agree with all of them. We made substantial revisions in light of the valuable comments of the reviewers and editors. All these revisions were appended to the manuscript in full accordance with the Sustainability’s guide for authors. Additionally, the flow of the manuscript was further improved to make the paper more informative and attractive for the readers. The editor and reviewers can find the revision details in the following sections. 

We believe that the paper’s quality was improved substantially thanks to these modifications based on the recommendations of reviewers and editors. We would like to thank you once again for your clear and constructive comments along with the fast review process.

Reviewer 2 Report

Comments and Suggestions for Authors

Abstract:

·         “The analysis results indicate that there are 18 459 interactions out of the 1859 linkages between LC and SDGs. Furthermore, 63 of 459 interactions 19 were found as "extremely important" interactions. Notably, LC principles L1, L9, and L8 are a stra-20 tegic trio to achieve SDGs in CI.”- This should be rewritten to indicate clear findings from the study. For instance, what are L1, L9, L8 to an average reader? Authors should replace this with what they stand for/denote.

·         “The study findings will provide”- The tense here should be corrected, will the study provide this or it is already provided through the contribution of the study.

·         The abstract says a literature review, focus group study and delphi was done. However, the title states the study adopts a quantitative approach. The authors should reconcile this statements.

Introduction

·         “The construction industry holds significance and consumes around half of the raw  materials presently extracted”- This statement is factual and should be supported.

·         “The SDGs consist of 17 main targets (including 169 goals) to address pov-37 erty, inequality, and bias were determined in 2015 (United Nations, 2018; Halkos & Gkam-38 poura, 2021; Goubran, 2019; Madurai Elavarasan et al., 2021). The construction industry 39 has a significant role in achieving the 2030 agenda for the SDGs due to its significant re-40 source and energy consumption (Galvez-Martos et al., 2018; Opoku, 2016; Dixon et al., 2018; Secher et al., 2018). Construction activities that focus on sustainability practices help to achieve SDGs” – The citations should be arranged with the earliest first.

·         “The review of the literature suggests that LC strongly intersects with environmental 60 sustainability, evident in waste reduction, promotion of value-added activities, mini-61 mized material waste, reduced working hours, and heightened safety measures (Nurlae-62 lah, 2023; Gao et al., 2023; Manzanar et al., 2022; Da Rocha et al., 2023)”- The citations should be arranged with the earliest first.

·         “Therefore, the purpose of the study is to find the synergy between LC principles and  SDGs according to the 2030 Agenda with a quantitative method (Delphi).”- How Is this only a quantitative method when the abstract also mentioned FGD?

·         The RQ should be revised. Why is synergy used repeatedly?

·         The RQ needs to be well defined. Why is there a question on the synergy between SDGs and LC philosophy according to the literature review when the following question states that the synergy between SDGs and LC philosophy based on Focus Group arose due to the lack of research in this field?

·         The introduction section should end with the contribution of the study, structure of the sections, and a brief description of the limitations of the study.

·         The  background of the study is not clear and does not provide the overall context of the problem. Specific problems needs to be addressed and highlighted.

·         The significance of the study also needs to be clearly stated. showing the relevance of the expected outcomes to practice (industry)

·         The research design should be clearly outlined in terms of the methodology (approach) and data collection methods and analysis towards the end of the introduction.

Section 2

·         “the key wastes being transportation, waiting, excess output, defects, inventory, motion, and excessive processing (Ohno & Bodek, 2019; Womack and Jones, 1997; Francis, &  Thomas,2020)”- The citations should be arranged with the earliest first.

·         Line 92-96- The citations should be arranged with the earliest first.

·         Kindly review the order of citations across the whole work. Also align to referencing style stipulated by the journal.

·         Critical review of related literature is not adequate

Methodology

·         “Quantitatively Research” in figure 1 should be revised

·         A separate figure detailing how the 39 articles were selected should be presented

·         Table 2- Why are some texts under The Synergy between LC and sustainability

·         underlined?

·         In figure 3-Correct, “Design questionnair” “Determain” “Distribute the QUestionnair”

·         Line 233 is not clear, kindly revise

·         Table 5 can be better presented.

Findings

·         linking of empirical findings to the literature review is not adequate

·         presentation and analysis of results in line with the research objectives can be better improved

·         Overall appearance and presentation needs to be revised as well.

Comments on the Quality of English Language

Extensive editing required

Author Response

(The authors gave the same response as above.)

Reviewer 3 Report

Comments and Suggestions for Authors

The study makes a significant contribution to knowledge by investigating the nexus between lean construction principles and UN SDGs. This is relevant for implementation purposes in all aspects of the SDGs. However, the presentation of the manuscript must improve before acceptance. The main points to address are:

Abstract

The abstract must be rewritten entirely to express the key findings and implications of the study. The abbreviations in the abstract must be written in full because this is the first attraction to the manuscript.

Introduction.

Line 28 states: "The construction industry holds significance and consumes around half of the raw materials presently extracted". This statement is too generic and can be explained further. Most of the sentences in this section need more critical analytical writing because they are disjointed and the flow of the sentences must improve. 

Literature review 

Table 1 has not identified recent studies on continuous improvement such as continuous cost improvement in construction: Theory and practice by Omotayo et al. (2022). 

The contents of Table 1 must be explained as part of an expanded literature review. This section is too brief considering the nature of the study. All the main principles must be discussed. 

Additionally, the SDGs must be discussed theoretically.

This can be done in just one page (briefly) since the study is a systematic review. 

Methodology

State the exclusion, inclusion and other criteria involved in the selection of the publications. What is the range in terms of years? More details must be produced in the methodology. 

Table 2 should be under a results heading.

Table 2 must have separate columns for LC and SDG, before having a column for the synergy. 

The presentation of the manuscript is confusing and seems rushed from these sections. 

Figure 3 has a typographical error "questionair"

Results and discussion

This section must include the results contained in the methodology (section 3)

Produce a new section after discussion. This should be on the implication/contributions of the findings for the construction sector and academia. 

what are the limitations of the study? 

References and presentation

The referencing style is wrong. Check the journal referencing specifications and adjust all references and citations. 

Comments on the Quality of English Language

Proofread the manuscript again. 

Author Response

(The authors gave the same response as above.)

Round 2

Reviewer 2 Report

Comments and Suggestions for Authors

Paper can be accepted

Reviewer 3 Report

Comments and Suggestions for Authors

This revised version of the manuscript implements all the suggested corrections. I would, therefore, recommend the manuscript for acceptance. 

Comments on the Quality of English Language

The manuscript must be revised extensively for typographical errors, sentence structure issues and references must be revised in the typesetting phase.